# Combination of Peroxisome Proliferator-Activated Receptor (PPAR) Alpha and Gamma Agonists Prevents Corneal Inflammation and Neovascularization in a Rat Alkali Burn Model

**DOI:** 10.3390/ijms21145093

**Published:** 2020-07-19

**Authors:** Yuji Nakano, Takeshi Arima, Yutaro Tobita, Masaaki Uchiyama, Akira Shimizu, Hiroshi Takahashi

**Affiliations:** 1Department of Ophthalmology, Nippon Medical School, Tokyo 113-8602, Japan; n-yuji@nms.ac.jp (Y.N.); takesuiii0714@nms.ac.jp (T.A.); y-tobita@nms.ac.jp (Y.T.); uchiyama@nms.ac.jp (M.U.); 2Department of Analytic Human Pathology, Nippon Medical School, Tokyo 113-8602, Japan; ashimizu@nms.ac.jp

**Keywords:** PPARα, PPARγ, combination of PPARα and PPARγ, cornea, alkali burn injury, neovascularization, inflammation, fibrotic changes

## Abstract

Peroxisome proliferator-activated receptor alpha (PPARα) and gamma (PPARγ) agonists have anti-inflammatory and anti-neovascularization effects, but few reports have tested the combination of PPARα and PPARγ agonists. In this study, we investigated the therapeutic effects of ophthalmic solutions of agonists of PPARα, PPARγ, and the combination in a rat corneal alkali burn model. After alkali injury, an ophthalmic solution of 0.05% fenofibrate (PPARα group), 0.1% pioglitazone (PPARγ group), 0.05% fenofibrate + 0.1% pioglitazone (PPARα+γ group), or vehicle (vehicle group) was topically instilled onto the rat’s cornea twice a day. After instillation, upregulation was seen of PPAR mRNA corresponding to each agonist group. Administration of agonists for PPARα, PPARγ, and PPARα+γ suppressed inflammatory cells, neovascularization, and fibrotic changes. In addition, the PPARγ agonist upregulated M2 macrophages, which contributed to wound healing, whereas the PPARα agonist suppressed immature blood vessels in the early phase. Administration of PPARα+γ agonists showed therapeutic effects in corneal wound healing, combining the characteristics of both PPARα and PPARγ agonists. The results indicate that the combination of PPARα and γ agonists may be a new therapeutic strategy.

## 1. Introduction

Transparency of the cornea is an essential factor for normal vision. An alkali burn can cause severe damage to the corneal surface, resulting in loss of its transparency [1,2]. Thus, suppression of scar formation and neovascularization during corneal wound healing is important for a good visual outcome.

Peroxisome proliferator-activated receptors (PPARs) are a group of nuclear receptors that belong to the steroid hormone receptor superfamily and act as transcription factors to control activation of genes [3,4]. PPARs have three isoforms—PPAR alpha (α), PPAR beta/delta (β/δ), and PPAR gamma (γ), and they are important factors in adipocyte differentiation and lipid metabolism [5]. Recent studies have shown their roles in lipid metabolism and also in the control of inflammation, angiogenesis, and fibrosis [6,7]. Furthermore, we previously showed anti-inflammatory and anti-neovascularization effects by PPARα and PPARγ agonists in a rat corneal alkali burn model [8,9]. 

Several studies have shown that the combination therapy of PPARα and PPARγ agonists is more effective than the use of either alone [10,11]. For example, addition of a PPARγ agonist to a PPARα agonist further reduces the expression of genes involved in insulin resistance, fatty acid synthesis, and fibrosis than when either agonist is applied alone [11]. However, as far as we know, little has been reported on the effects of combined PPARα and PPARγ agonists in the field of ophthalmology. In the present study, we compared the anti-inflammatory, anti-neovascularization, and anti-fibrosis effects of fenofibrate, a selective agonist of PPARα, pioglitazone, a selective agonist of PPARγ, and their combination in a rat corneal alkali burn model.

## 2. Results

### 2.1. Corneal Wound Healing after Alkali Burn

Effects of each PPAR agonist ophthalmic solution were examined by macroscopic observation (Figure 1) and histological analysis using hematoxylin and eosin (HE) staining (Figure 2). In the macroscopic photographs, blood vessels in the iris disappeared, and the corneal epithelial defect was seen as a circle line immediately after the injury (Figure 1a). On day 7, the cornea lost transparency, and bleeding in the anterior chamber occurred in the vehicle group (Figure 1b). On the other hand, the corneal opacity and neovascularization were less obvious in all PPAR instillation groups (Figure 1c–e). Furthermore, corneal opacity was minimal in the cornea in the PPARα+γ group (Figure 1e). Microscopically, various infiltrating inflammatory cells, including macrophages and neutrophils, were seen in the peripheral cornea at 6 h after the alkali burn in all groups (Figure 2b,e,h,k). By day 7, the infiltration of inflammatory cells had decreased in the peripheral cornea (Figure 2c,f,i,l), and these cells had migrated to the center of the cornea (Figure 2a,d,g,j). In addition, neovascularization was observed in the corneal limbus (Figure 2c,f,i,l). All PPAR groups showed a lesser degree of infiltrating inflammatory cells and neovascularization compared to the vehicle group on day 7.

### 2.2. PPARα and PPARγ mRNA Expression in the Cornea

Real-time reverse transcription polymerase chain reaction (RT-PCR) was performed to investigate mRNA expression levels of PPARα and PPARγ in the burned cornea at 6 h after instillation of each PPAR (Figure 3). PPARα and PPARα+γ treatment increased PPARα mRNA expression compared with the vehicle and PPARγ treatment (Figure 3a). PPARγ and PPARα+γ treatment also increased PPARγ mRNA expression in comparison to the vehicle group (Figure 3b).

### 2.3. Anti-Inflammatory Effects of the PPAR Agonist Ophthalmic Solution

To investigate infiltrating neutrophils, naphthol AS-D chloroacetate esterase (EST) staining was performed. EST-positive neutrophils were noted in the corneal limbus at 6 h and increased by day 1 after the alkali burn (Figure 4). On day 1, the number of neutrophils peaked in the injured cornea in all groups. The ophthalmic solution of PPARγ and PPARα+γ agonists decreased the degree of infiltrating neutrophils compared to vehicle treatment at 6 h. Instillation of all PPAR agonists reduced the number of neutrophils in the cornea on day 1 (Figure 4e). 

Real-time RT–PCR analysis was performed at 6 h after alkali injury to assess the expression of genes associated with proinflammatory cytokines, including interleukin-1β (IL-1β), IL-6, nuclear factor-kappa B (NF-κB), nuclear factor of kappa light polypeptide gene enhancer in B-cells inhibitor alpha (IκB-α), and tumor necrosis factor-α (TNF-α). All PPAR treatments equally suppressed expression of IL-1β and IL-6 mRNA (Figure 5a,b). Instillation of PPARγ and PPARα+γ agonists also reduced the mRNA expression of TNF-α and NF-κB in comparison with vehicle instillation (Figure 5c,d). On the other hand, only PPARα treatment increased the mRNA expression of IκB-α compared with vehicle treatment (Figure 5e). 

Therefore, we performed immunohistochemical analysis to investigate expression of NF-κB and IκB-α in corneal epithelium cells at 6 h and four days after the injury (Figure 6). NF-κB and IκB-α staining was observed at a similar location in the corneal epithelium. At 6 h after the injury, NF-κB expression was high and localized in the nuclear area (Figure 6a), and IκB-α was slightly expressed in the cytoplasm in the vehicle group (Figure 6b). PPARα instillation significantly enhanced IκB-α expression (Figure 6d). In the PPARγ and PPARα+γ treatment groups, less NF-κB expression was present than in the vehicle group and the PPARα group (Figure 6e,g). At four days after the injury, epithelial proliferation was observed in the vehicle group, and NF-κB and IκB-α expression was localized in nearly identical cells (Figure 6i,j). After PPARα treatment, NF-κB was mainly expressed in the cytoplasm, and IκB-α expression was localized in the nuclear area (Figure 6k,l). PPARγ instillation reduced NF-κB expression, and little IκB expression was seen in comparison with instillation of vehicle or PPARα (Figure 6m,n). PPARα+γ treatment strongly decreased NF-κB expression compared to the other three groups (Figure 6o,p).

We also investigated macrophage infiltration using immunohistochemical analysis with CD68 antibody (ED-1) and CD163 antibody (ED-2) (Figure 7). In the vehicle group, the number of ED-1-positive macrophages increased in the peripheral cornea at 6 h after the injury, and peaked on day 1, similar to EST-positive neutrophils. PPARγ and PPARα+γ treatment significantly suppressed ED-1-positive macrophages on day 1 compared with vehicle treatment (Figure 7c). On the other hand, ED-2-positive cells increased in the PPARγ and PPARα+γ groups on day 4 (Figure 7f).

### 2.4. Suppression of Neovascularization by Instillation of PPAR Agonists

Next, immunohistochemical analysis of nestin and aminopeptidase P (JG12) was conducted to investigate neovascularization. Nestin-positive newly formed endothelial cells were observed in the peripheral cornea on day 4 and day 7 after the alkali burn (Figure 8a–c). Although each PPAR agonist ophthalmic solution reduced nestin-positive endothelial cells on day 7, the PPARα agonist suppressed neovascularization from the early phase on day 4 (Figure 8c). JG-12-positive capillary lumens were observed on day 4, and these were increased by day 7 after the corneal alkali injury in the vehicle group. The different PPAR treatments equally suppressed the increase in capillary lumens by day 7 (Figure 8f).

Real-time RT–PCR analysis was performed in the cornea at 6 h after the alkali burn to assess the expression of genes associated with neovascularization, including vascular endothelial growth factor-A (VEGF-A), angiopoietin-1 (Ang-1), and Ang-2 (Figure 9a–c). A significant difference was present in the mRNA expression levels of VEGF-A between each PPAR treatment group and the vehicle group (Figure 9a). Furthermore, instillation of all PPAR agonists significantly downregulated mRNA expression of Ang-2 at 6 h (Figure 9c). No significant differences were found in Ang-1 mRNA expression among all groups (Figure 9b). The expression level of Ang-2 mRNA was comparatively higher than that of Ang-1.

### 2.5. Anti-Fibrotic Effects of PPAR Agonist Instillation

Immunohistochemical analysis of collagen type III was performed to evaluate fibrotic changes at seven days after the alkali injury (Figure 10a–f). On day 7 after injury, we noted the accumulation of collagen type III in the corneal stroma (Figure 10a–d). All PPAR instillation groups exhibited a lower volume of collagen type III (Figure 10b–d) and a decrease in the area of collagen type III expression compared to the vehicle (Figure 10e). Interestingly, percentages of collagen type III in the corneal regions were significantly lower in the PPARα+γ group compared to either PPAR alone group. We also investigated expression of mRNA for transforming growth factor-β (TGF-β), which is involved in the fibrotic reaction. Instillation of PPARγ and PPARα+γ agonists reduced the mRNA expression of TGF-β1 in comparison with vehicle instillation (Figure 10f).

## 3. Discussion

Corneal alkaline injury is one of the most severe ophthalmic emergencies and requires prompt treatment because it usually results in a serious outcome accompanied with corneal neovascularization and opacity [2,12]. Topical therapy with PPAR agonists is expected to be a new treatment strategy for alkali-burned corneas [13]. Our present study demonstrated the following: (1) PPARα and PPARγ agonist treatment upregulated the mRNA expression of PPARα and PPARγ, respectively. In addition, PPARα+γ agonist instillation increased the mRNA expression of both PPARα and PPARγ. (2) Administration of all topical PPAR agonists suppressed infiltrating inflammatory cells in the early phase after the alkali burn and mRNA expression of proinflammatory cytokines. On the other hand, PPARγ and PPARα+γ significantly increased the number of M2 macrophages on day 4. (3) Instillation of all PPAR agonists prevented neovascularization by reducing VEGF-A and Ang-2 expression.

The strong inflammation induced by alkali injury to the mouse cornea leads to corneal epithelial defects and upregulation of various cytokines such as IL-1β and IL-6, which play important roles in wound healing after an alkali burn [14]. In the present study, the PPARα agonist and PPARγ agonist significantly suppressed these cytokines, suggesting that both agonists induced significant anti-inflammatory effects in the alkali-burned cornea.

In addition, the PPARγ agonist significantly reduced mRNA expression of TNF-α and NF-κB, whereas the PPARα agonist increased mRNA expression of IκB-α. TNF-α is involved in the inflammatory pathway, and thus decreased TNF-α mRNA expression by PPARγ treatment seems effective in corneal wound healing after alkali injury. Furthermore, NF-κB plays a major role in regulation of the expression of a number of genes involved in cell growth, inflammation, and apoptosis [15,16]. Both the PPARα agonist and PPARγ agonist induce IκB [17,18,19,20], which is the inhibitory protein of NF-κB. IκB inhibits the effect of NF-κB in the cytoplasm by blocking NF-κB binding to DNA [16]. In this study, only PPARα increased IκΒ-α expression at 6 h after the alkali injury, suggesting downregulation of NF-κB. Immunohistochemical analysis of NF-κB and IκB-α showed that both NF-κB and IκB-α were highly expressed in the cytoplasm after PPARα instillation in the early phase (Figure 6). On the other hand, immunostaining revealed that PPARγ instillation strongly reduced NF-κB expression, but rarely activated IκB-α expression. These results suggested that PPARγ has an anti-inflammatory effect by reducing NF-κΒ expression without upregulation of IκB. Thus, PPARα agonists and PPARγ agonists may have a different mechanism of anti-inflammatory effects.

Macrophages are classified as either classically activated (M1) or alternatively activated (M2) cells and can change their subtype [21]. M1 macrophages are tissue injury-type macrophages involved in the development of inflammation [22]. Conversely, M2 macrophages play immunoregulatory and immunosuppressive roles. M2 macrophages reduce inflammation by producing anti-inflammatory factors, and they affect tissue remodeling and repair [23]. Activation of PPARγ polarizes monocytes to become M2 macrophages in the circulating blood [24]. Our previous study also revealed that a PPARγ agonist was one factor underlying the M1/M2 balance [8]. Our current study showed an increase in M2 macrophages after PPARγ treatment on day 4, suggesting that administration of the PPARγ agonist played another critical role in tissue remodeling and wound healing.

Neovascularization is mediated by molecular factors. VEGF-A is an important molecule for angiogenesis [25]. Ang-1 and Ang-2 are the ligands of the Tie2 receptor. Ang-1 is the major ligand for the Tie2 receptor and may induce maturation and stabilization of developing blood vessels [26]. On the other hand, Ang-2 has opposite effects of Ang-1 and may cause destabilization of developing blood vessels [27]. Although blood vessel formation is regulated by VEGF along with Ang-1 and Ang-2, neither Ang-1 nor Ang-2 alone can promote angiogenesis [28]. The combination of Ang-1 and VEGF increases macroscopically evident perfusion of corneal neovascular structures. In contrast, the combination of Ang-2 and VEGF promotes neovascularization [29]. PPARα and PPARγ agonists prevent neovascularization in the eye by suppressing inflammatory factors and neovascularization factors [9,30,31]. In our present study, both the PPARα and PPARγ agonists reduced expression of VEGF and Ang-2 mRNA at 6 h after the alkali burn. The number of immature lumens on day 4 and mature blood vessels on day 7 after instillation of PPARα and PPARγ agonists was lower than after vehicle instillation. These results indicated that neovascularization was prevented by suppressing the levels of VEGF and Ang-2 mRNA expression at 6 h after the alkali injury. Regarding the PPARα agonist, immature juvenile blood vessels were suppressed from the early phase.

The normal cornea mainly consists of collagen type I, the expression of which is tightly regulated, that maintains transparency [32]. Collagen type III plays an important role in corneal wound healing, and unregulated accumulation of collagens can result in an opaque cornea [33]. The present study showed deposition of collagen type III in the cornea at seven days after the alkali injury and that inflammation suppressed by PPARα or PPARγ instillation resulted in less scar formation. TGF-β is also an important profibrotic element, especially in fibrosis [34,35]. Rosglitazone, which is a PPARγ agonist, suppresses TGF-β1 expression and prevents conjunctival fibrosis after glaucoma filtration surgery [36]. In our current study, PPARγ treatment suppressed fibrotic changes in the corneal stroma, which may be induced by anti-TGF-β effects of PPARγ.

A few studies have reported the efficiency of combined PPARα and PPARγ agonist therapy for diabetes and dyslipidemia [10,11]. The combination of half doses of fenofibrate and pioglitazone improves nonalcoholic steatohepatitis-related disturbances, similar to a full dose of fenofibrate [11]. In another report, pre-treatment with fenofibrate, pioglitazone, and their combination in rats with cerebral ischemia improves inflammatory and apoptotic markers, such as TNF-α [37]. However, little has been reported on the effects of the combination of PPARα and PPARγ agonist therapy in the field of ophthalmology. In this study, subtle differences were observed in the anti-inflammatory and anti-angiogenic effects of each PPAR agonist. Fenofibrate, a PPARα agonist, suppressed angiogenesis from the early phase, whereas pioglitazone, a PPARγ agonist, enhanced the anti-inflammatory effect by upregulating M2 macrophages. Furthermore, the combination of fenofibrate and pioglitazone significantly prevented corneal accumulation of collagen type III compared to either alone. NF-κB expression in the corneal epithelium was strongly reduced by the combination of PPARα and PPARγ at four days after the injury. A different mechanism involved in NF-κB and IκB-α expression after fenofibrate and pioglitazone treatment may contribute to the significant downregulation of NF-κB, thereby suppressing scar formation. Therefore, instillation of a drug combination may be more effective than a single PPAR agonist for corneal alkali burn. Recently, a few reports have been published on a PPARα/γ dual agonist [38,39]. Kaur et al. reported that trans-acrylic acid derivatives have binding affinity for PPARα and PPARγ and significantly reduce the plasma glucose level and total cholesterol level in diabetic rats compared to pioglitazone [39]. Further studies are needed to investigate the therapeutic effect of a dual agonist following corneal alkali burn.

## 4. Materials and Methods

### 4.1. Animals and Ethics Statement

Eight-week-old male *Wistar* rats (*n* = 8 per group/per endpoint; Sankyo Laboratory Service, Tokyo, Japan) were used for all experiments in the present study.

All animal experiments were performed in accordance with the Association for Research in Vision and Ophthalmology (ARVO) Statement for the Use of Animals in Ophthalmic and Vison Research. The Experimental Animal Ethics Review Committee of Nippon Medical School, Tokyo approved the current research procedures (approval number: 2019-15, 1 April 2019).

### 4.2. Alkali Burn Model and TREATMent with PPAR Agonist Ophthalmic Solution

The corneal alkali burn was made by placing a circular piece of filter paper, 3.2 mm in diameter, which was soaked in 1 M NaOH, on the central cornea for 1 min under general isoflurane anesthesia. After alkali exposure, the cornea was rinsed immediately with 40 mL physiological saline. The procedure was performed in the right eye of each rat.

Immediately after the above treatment, vehicle solution (vehicle/control group), 0.05% fenofibrate solution (PPARα group), 0.1% pioglitazone solution (PPARγ group), or 0.05% fenofibrate + 0.1% pioglitazone solution (PPARα+γ group) was applied to the ocular surface of each rat. Topical administration was continued in each group twice a day until the endpoint. Macroscopic photographs of each group were obtained at each time point. Each eye drop solution was prepared as follows. Vehicle solution was made using 0.1 mL polyoxyethylene sorbitan monooleate (Wako Pure Chemical Industries, Osaka, Japan) and 100 mL NaCl-based phosphate-buffered saline (0.01 M; pH 7.4), which was prepared with disodium hydrogen phosphate dodecahydrate (232 g), sodium dihydrogen phosphate dihydrate (23.7 g), and distilled water (4000 mL). The 0.05% fenofibrate solution was made by adding 10 mg fenofibrate (Wako Pure Chemical Industries) to 20 mL of vehicle solution. The 0.1% pioglitazone solution was made by adding 20 mg pioglitazone (Wako Pure Chemical Industries) to 20 mL of vehicle solution. The combination of 0.05% fenofibrate and 0.1% pioglitazone was made by adding 10 mg fenofibrate and 20 mg pioglitazone to 20 mL of vehicle solution. All solutions were kept at 4 °C.

Rats were euthanized at each time point by exsanguination under general isoflurane anesthesia. The eyeballs were enucleated for histological and immunohistochemical analysis and real-time RT-PCR. The corneal tissues for RT-PCR were immediately placed in RNAlater solution (Life Technologies, Carlsbad, CA, USA).

### 4.3. Histological and Immunohistochemical Analysis

At 6 h and on days 1, 4, and 7 after the alkali burn, the eyes were enucleated for histological and immunohistochemical analysis (*n* = 8 per time point), fixed in 10% buffered formalin, and embedded in paraffin for light microscopic analysis. Deparaffinized tissues were stained with HE for histopathological examination. EST staining was performed to detect infiltrating neutrophils [40].

Primary antibodies used for immunohistochemical analysis included: polyclonal rabbit anti-NF-κB/P65 (Santa Cruz Biotechnology, Dallas, TX, USA), monoclonal mouse anti-IκB-α (Santa Cruz Biotechnology), monoclonal mouse anti-rat ED-1 (BMA, Nagoya, Japan; to detect infiltrating macrophages), monoclonal mouse anti-rat ED-2 antibody (BMA; to detect M2 macrophages), monoclonal mouse anti-nestin (Nestin; Merck Millipore, Darmstadt, Germany; to detect endothelial cells of immature blood vessels [41]), monoclonal mouse anti-JG12 (Thermo Fisher Scientific, Tokyo, Japan), which is specifically expressed by endothelial cells of blood vessels [42], and polyclonal goat anti-type III collagen (Southern Biotech, Birmingham, AL, USA).

For all immunohistochemical analyses, paraffin-embedded tissues that were fixed in 10% buffered formalin were used. The tissues were stained using a standard avidin-biotin-peroxidase complex technique. The numbers of immunohistochemically positive cells were counted by one researcher, and the average of five fields in the cornea at 400× magnification was calculated. Cornea areas in three fields of the center cornea at a magnification of 200× were measured using ImageJ software.

### 4.4. Real-Time RT-PCR

To examine the mRNA expression levels of PPARα, PPARγ, IL-1β, IL-6, NF-κB, IκB, TNF-α, TGF-β1, VEGF-A, Ang-1, and Ang-2, we used real-time RT–PCR (*n* = 8 per time point). Total RNA was extracted from corneas using ISOGEN II (Nippon Gene, Tokyo, Japan) in accordance with the manufacturer’s protocol. To measure the RNA concentration and ensure purity (A260/A280), a NanoDrop ND-1000 V3.2.1 Spectrophotometer (NanoDrop Technologies, Wilmington, DE, USA) was used. cDNA libraries were synthesized from total RNA using the High Capacity cDNA Reverse Transcription kit (Thermo Fisher Scientific) in accordance with the manufacturer’s protocol. Target genes were amplified (2 min at 50 °C, 10 min at 95 °C, and 45 cycles of denaturation at 95 °C for 15 s and annealing at 60 °C for 60 s) using the QuantStudio^TM^ 3 Real-Time PCR System (Thermo Fisher Scientific), THUNDERBIRD SYBR qPCR Mix (TOYOBO, Osaka, Japan), and specific primers. Normalized values for mRNA expression were calculated as the relative quantity of the housekeeping gene, β-actin. The primers used in this experiment are described below (Table 1).

### 4.5. Statistical Analyses

All results are expressed as the mean ± standard error. Different groups were compared using one-way analysis of variance followed by the Tukey–Kramer post-analysis test. A *p*-value less than 0.05 was considered statistically significant. All analyses were calculated using GraphPad Prism software (Version 8.4.2, GraphPad Software, San Diego, CA, USA).

## 5. Conclusions

The combination ophthalmic solution of fenofibrate and pioglitazone suppressed inflammation, neovascularization, and fibrotic changes by a dual therapeutic effect of the PPARα and PPARγ agonists in the alkali-burned cornea, suggesting a new candidate therapeutic strategy for corneal disease.

## Figures and Tables

**Figure 1 ijms-21-05093-f001:**
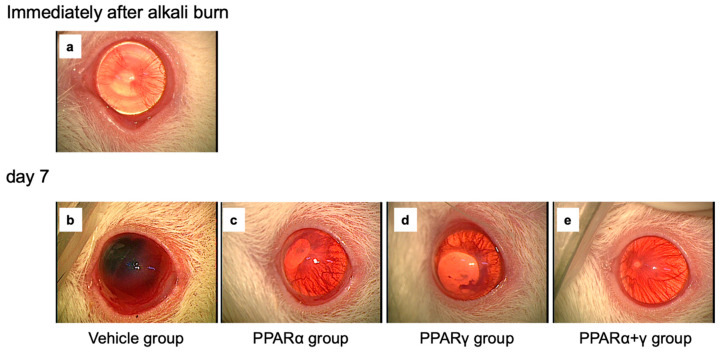
Anterior segment photographs of the cornea in each group after alkali burn. In the cornea immediately after the burn, blood vessels of the iris disappeared, and a circular epithelial defect line was observed (**a**). Anterior chamber bleeding occurred in the vehicle group (**b**). The central opacity and neovascularization of the cornea were less severe in all peroxisome proliferator-activated receptor (PPAR) groups than in the vehicle group on day 7 (**c**–**e**). In the PPARα+γ group in particular, the corneal opacity was quite slight (**e**).

**Figure 2 ijms-21-05093-f002:**
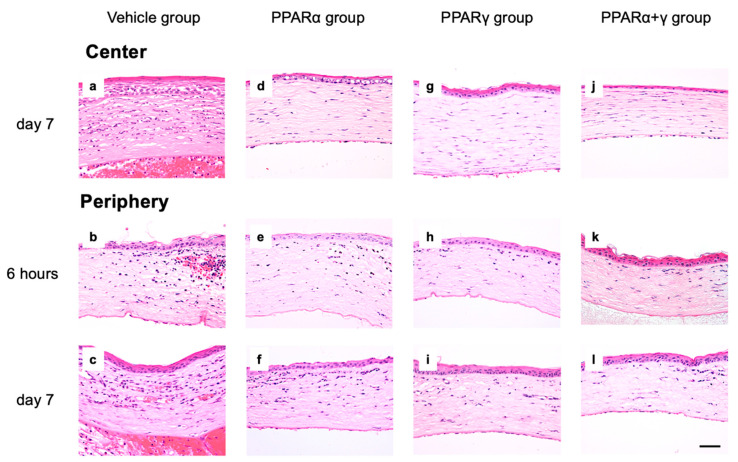
Histological analysis following hematoxylin and eosin (HE) staining. The central and peripheral cornea of the vehicle group (**a**–**c**), PPARα group (**d**–**f**), PPARγ group (**g**–**i**), and PPARα+γ group (**j**–**l**) at 6 h and day 7. Bar, 50 μm. Various inflammatory cells infiltrated from the peripheral cornea at 6 h and migrated to the central cornea by day 7. Furthermore, luminal structures that appear to be blood vessels were seen in the peripheral cornea on day 7. In all PPAR instillation groups, low numbers of infiltrating cells and luminal structures, and corneal stromal edema were seen.

**Figure 3 ijms-21-05093-f003:**
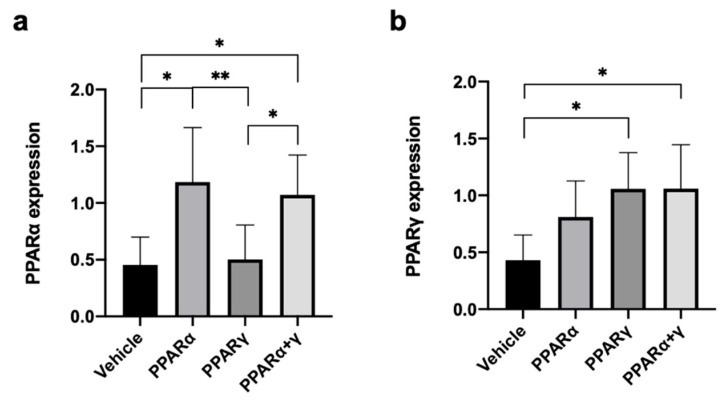
Expression of PPARα and PPARγ mRNA. At 6 h, real-time RT-PCR showed upregulation of mRNA expression levels of PPARα in the alkali-burned cornea after PPARα and PPARα+γ treatment compared with the vehicle and the PPARγ group (**a**). In addition, mRNA expression levels of PPARγ were increased in the PPARγ and PPARα+γ groups in comparison to the vehicle group (**b**). Data are presented as the mean ± standard error (*n* = 8 samples/group). ***p* < 0.01, **p* < 0.05.

**Figure 4 ijms-21-05093-f004:**
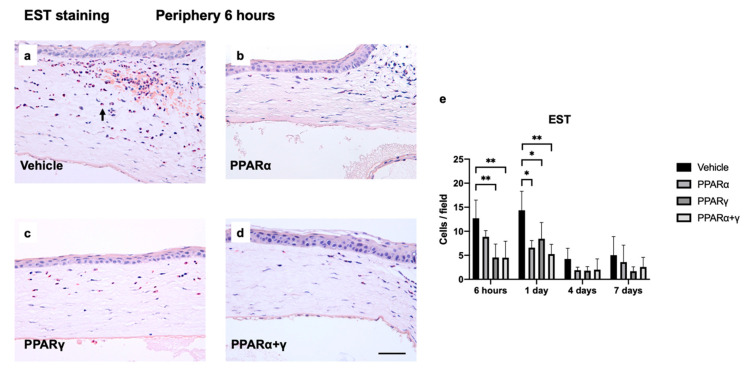
Esterase (EST) staining was performed to evaluate infiltrating cells in the rat cornea after alkali injury. Immunohistochemical images of the corneal limbus at 6 h are shown (**a**–**d**). Bar, 50 μm. The black arrow indicates EST-positive cells (**a**). PPARγ and PPARα+γ treatment reduced the number of neutrophils in the peripheral cornea at 6 h. Bar chart of the number of EST-positive cells shows a statistically significant difference between vehicle and PPAR treatments on day 1 (**e**). At 6 h after alkali burn, instillation of PPARγ and PPARα+γ agonists significantly suppressed neutrophils compared with vehicle and PPARα instillation (**e**). Data are presented as the mean ± standard error (*n* = 8 samples/group). ***p* < 0.01, **p* < 0.05.

**Figure 5 ijms-21-05093-f005:**
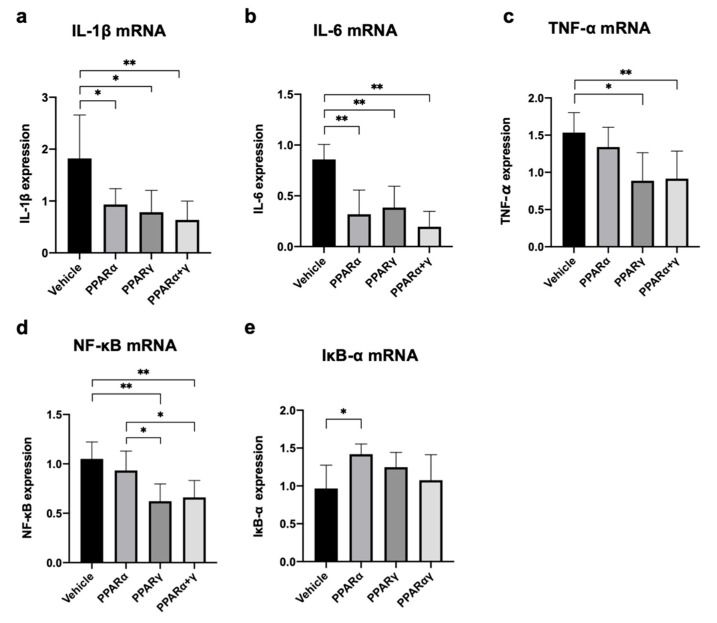
Expression of mRNA for proinflammatory cytokines in the cornea at 6 h after alkali burn. We measured mRNA expression levels of IL-1β (**a**), IL-6 (**b**), TNF-α (**c**), NF-κB (**d**), and IκB-α (**e**). Treatment with all PPAR agonists equally suppressed expression of IL-1β and IL-6 mRNA (**a** and **b**). Instillation of PPARγ and PPARα+γ agonists also reduced the mRNA expression of TNF-α and NF-κB in comparison with vehicle instillation (**c**), (**d**). On the other hand, only PPARα treatment increased the mRNA expression of IκB-α compared with vehicle treatment (**e**). Data are presented as the mean ± standard error (*n* = 8 samples/group). ***p* < 0.01, **p* < 0.05.

**Figure 6 ijms-21-05093-f006:**
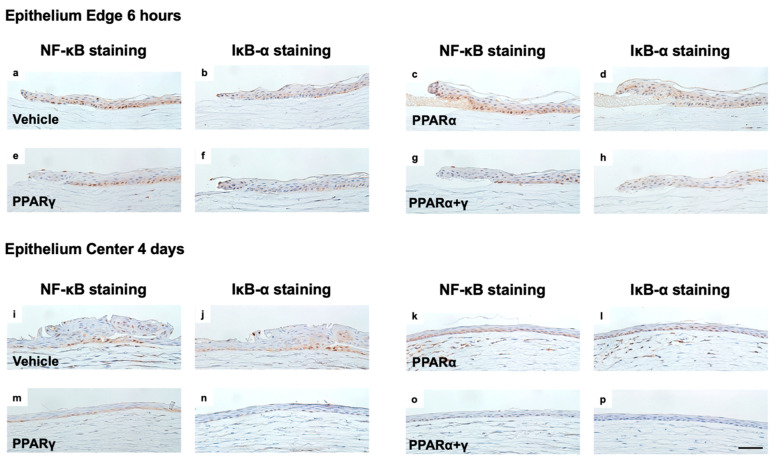
Immunostaining for NF-κB and IκΒ-α in epithelium cells in each group at 6 h and four days after the injury. Representative sections at 6 h (**a**–**h**) and four days (**i**–**p**) are shown. At 6 h after the injury, NF-κB was strongly expressed in the cell nuclei in the vehicle group (**a**), whereas it was expressed at low levels in the nuclei in the PPARα group (**c**). IκB-α was expressed in the cytoplasm after PPARα instillation (**d**). In the PPARγ and PPARα+γ groups, less NF-κB expression was seen compared to the vehicle group and the PPARα group (**e**), (**g**). The PPARα+γ group exhibited a lesser degree of NF-κB-positive inflammatory cell infiltration compared to the other three groups on day 4 after the injury (**i**–**p**). Bar, 50 μm (**a**–**p**).

**Figure 7 ijms-21-05093-f007:**
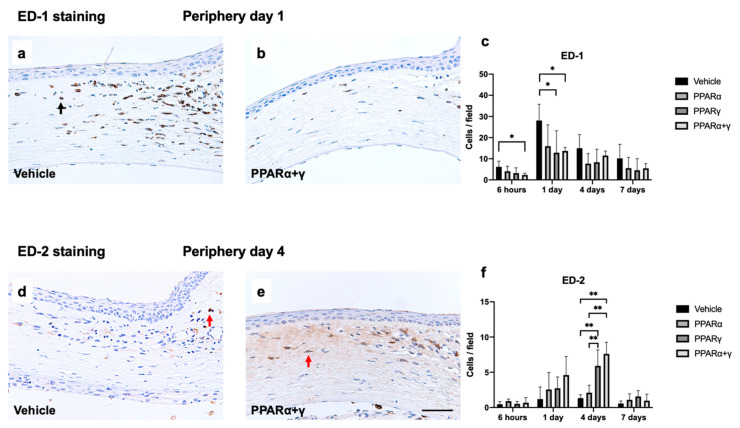
Infiltration of macrophages in the cornea after alkali injury. Representative photomicrographs of infiltrating macrophages in the corneal limbus after instillation of the vehicle (**a**) and PPARα+γ (**b**) on day 1 showed prominent infiltration of ED-1-positive macrophages on day 1 in the vehicle group, but significantly suppressed infiltration in the PPARα+γ group (**c**). Black arrow indicates an ED-1-positive cell (**a**). PPARγ and PPARα+γ treatment significantly suppressed ED-1-positive macrophages on day 1 compared with vehicle treatment (**c**). Data are presented as the mean ± standard error (*n* = 8 samples/group). **p* < 0.05. Although few ED-2-positive M2 macrophages were present in the peripheral cornea on day 4 after alkali burn in the vehicle group (**d**), PPARα+γ treatment upregulated M2 macrophages (**e**). Red arrows indicate ED-2-positive cells (**d**), (**e**). Bar chart of the number of ED-2-positive cells shows a statistically significant difference between the vehicle and PPARγ groups, similar to the PPARα+γ instillation group (**f**). Bar, 50 μm (**a**), (**b**), (**d**), (**e**). Data are presented as the mean ± standard error (*n* = 8 samples/group). ***p* < 0.01, **p* < 0.05.

**Figure 8 ijms-21-05093-f008:**
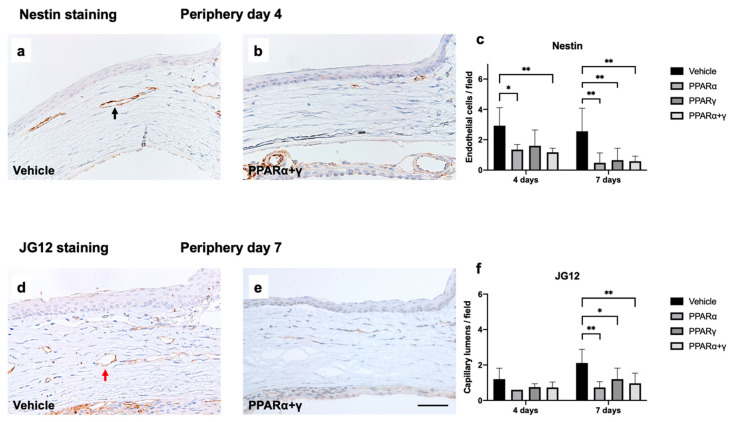
Immunohistochemical analysis of nestin (**a**–**c**) and JG12 (**d**–**f**) in the vehicle (**a**), (**d**) and PPARα+γ (**b**), (**e**) groups. Bar, 50 μm. Nestin-positive endothelial cells were observed during the early phase in the corneal limbus (**a**), (**b**). The black arrow indicates nestin-positive endothelial cells of a new blood vessel (**a**). Bar charts of the number of nestin-positive cells indicate that all PPAR agonist ophthalmic solutions reduced nestin-positive endothelial cells in the later phase (day 7). In the early phase (day 4), newly formed blood vessels were significantly lower in the PPARα and PPARα+γ groups than in the vehicle group (**c**). Data are presented as the mean ± standard error (*n* = 8 samples/group). ***p* < 0.01, **p* < 0.05. JG-12-positive capillary lumens were observed on day 7 (**d**), (**e**). Red arrow indicates a JG12-positive cell (**d**). The number of capillaries increased by day 7 after corneal alkali injury in the vehicle group. Each PPAR treatment significantly suppressed the increase in capillary lumens by day 7 compared to the vehicle group (**f**). Data are presented as the mean ± standard error (*n* = 8 samples/group). ***p* < 0.01, **p* < 0.05.

**Figure 9 ijms-21-05093-f009:**
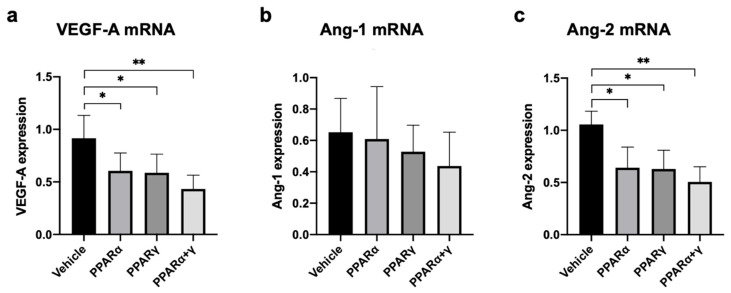
Expression of mRNA for molecules involved in neovascularization. Real-time RT–PCR analysis was performed to investigate the level of VEGF-A mRNA (**a**), Ang-1 mRNA (**b**), and Ang-2 mRNA (**c**) in the cornea at 6 h after alkali burn. Instillation of all PPAR agonists suppressed mRNA expression of VEGF-A (**a**). All PPAR treatments significantly reduced mRNA expression of Ang-2 at 6 h (**c**). In contrast, no significant differences were seen in the degree of Ang-1 mRNA expression between each group (**b**). The mRNA expression level of Ang-2 mRNA was relatively higher than that of Ang-1 mRNA. Data are presented as the mean ± standard error (*n* = 8 samples/group). ***p* < 0.01, **p* < 0.05.

**Figure 10 ijms-21-05093-f010:**
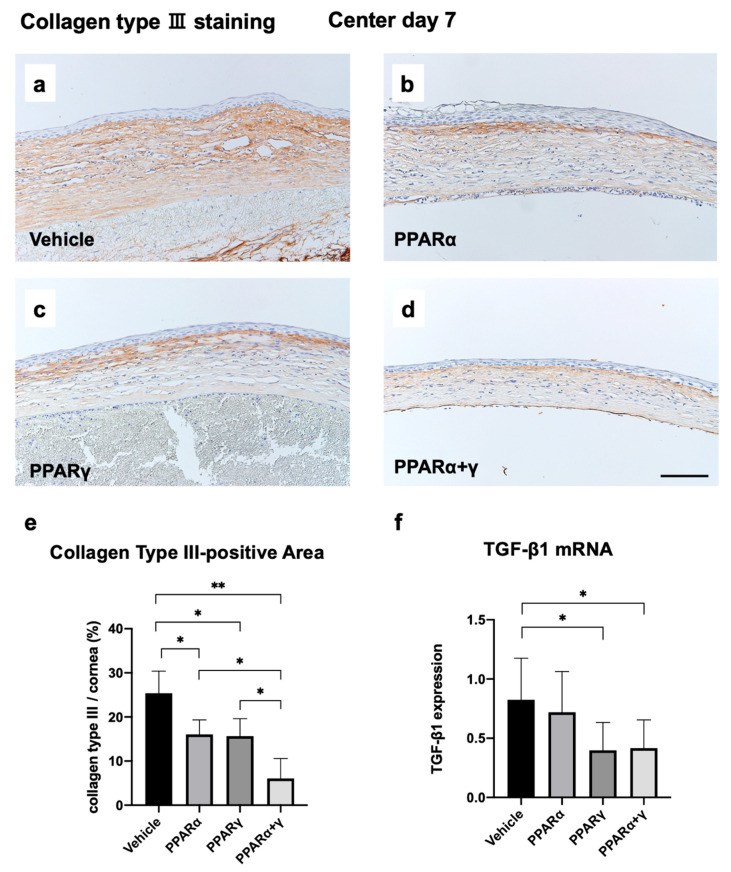
Immunohistochemical analysis of collagen type III to evaluate fibrotic changes at seven days after the alkali injury (**a**–**f**). Collagen type III was observed in the injured area during the healing process in the vehicle group (**a**). Instillation of each PPAR reduced collagen type III expression (**b–d**). Bar chart of percentages of collagen type III expression area/cornea area showed that the PPARα+γ group had a significantly lower percentage of collagen type III compared to other groups (**e**). Real-time RT-PCR showed marked suppression of TGF-β mRNA expression in the burned cornea after PPARγ or PPARα+γ agonist treatment compared with the vehicle group (**f**). Bar, 100 μm. Data are presented as the mean ± standard error (*n* = 8 samples/group). ***p* < 0.01, **p* < 0.05.

**Table 1 ijms-21-05093-t001:** Primer pairs used in this study.

Gene	Forward Primer Sequence (5′-3′)	Reverse Primer Sequence (5′-3′)
β-actin	GCAGGAGTACGATGAGTCCG	ACGCAGCTCAGTAACAGTCC
PPARα	TCGTGGAGTCCTGGAACTGA	GAGTTACGCCCAAATGCACC
PPARγ	GCGAGGGCGATCTTGACA	ATGCGGATGGCCACCTCTTT
IL-1β	TACCTATGTCTTGCCCGTGGAG	ATCATCCCACGAGTCACAGAGG
IL-6	GTCAACTCCATCTGCCCTTCAG	GGCAGTGGCTGTCAACAACAT
NF-κΒ	TGGACGATCTGTTTCCCCTC	TCGCACTTGTAACGGAAACG
IκB-α	TGACCATGGAAGTGATTGGTCAG	GATCACAGCCAAGTGGAGTGGA
ΤNF-α	AAATGGGCTCCCTCTCATCAGTTC	TCTGCTTGGTGGTTTGCTACGAC
TGF-β1	TGGCCAGATCCTGTCCAAAC	GTTGTACAAAGCGAGCACCG
VEGF-A	GCAGCGACAAGGCAGACTAT	GCAACCTCTCCAAACCGTTG
Ang-1	CACCGTGAGGATGGAAGCCTA	TTCCCAAGCCAATATTCACCAGA
Ang-2	CTTCAGGTGCTGGTGTCCA	GTCACAGTAGGCCTTGACCTC

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
