# Peer review of "Combination of Peroxisome Proliferator-Activated Receptor (PPAR) Alpha and Gamma Agonists Prevents Corneal Inflammation and Neovascularization in a Rat Alkali Burn Model"

_ijms, 2020, doi:10.3390/ijms21145093_

Round 1

Reviewer 1 Report

Yuji Nakano et al have submitted the manuscript “Combination of peroxisome proliferator-activated receptor (PPAR) alpha and gamma agonists prevents corneal inflammation and neovascularization in a rat alkali burn model.” The paper seeks to address the combination of fenofibrate and pioglitazone suppressed inflammation and neovascularization by a dual therapeutic effect of the PPARα and PPARγ agonists in the  alkali-burned cornea. The authors hypothesized that the anti-inflammatory and anti-neovascularization effects of  combination of PPARα and PPARγ would prevent corneal damage in a rat corneal alkali burn model. The authors perform a series of studies, which suggest that the cmbination prevents corneal inflammation and neovascularization. However, it is difficult to judge whether the study answer additive synergy of PPARα and PPARγ. The combination treatment is obviously successful in anterior segment photographs and histological analysis of the cornea, but any markers in your study would not show differences between the combination and each monotherapy. Thus, it is not possible to recommend the combination over PPAR monotherapy because enhanced monotherapy might be more effective.Please try to make the hypothesis clear and show the additive effect of the PPARs.

Author Response

We are grateful for the opportunity to revise and resubmit our paper (ijms-849803) entitled, “Combination of peroxisome proliferator-activated receptor (PPAR) alpha and gamma agonists prevents corneal inflammation and neovascularization in a rat alkali burn model”, and would like to thank you for your helpful comments.

Attached to this letter is a revised version of our manuscript that shows the tracked changes. In addition, we have separately listed our point-by-point responses to the comments in the text that follows.

We would like to thank you for your constructive comments. After incorporating the suggested changes, this has greatly improved our paper, thereby making this a much stronger manuscript. Please convey our thanks to you for all of your helpful input.

Our responses to the comments are as follows:

Yuji Nakano et al have submitted the manuscript “Combination of peroxisome proliferator-activated receptor (PPAR) alpha and gamma agonists prevents corneal inflammation and neovascularization in a rat alkali burn model.” The paper seeks to address the combination of fenofibrate and pioglitazone suppressed inflammation and neovascularization by a dual therapeutic effect of the PPARα and PPARγ agonists in the  alkali-burned cornea. The authors hypothesized that the anti-inflammatory and anti-neovascularization effects of  combination of PPARα and PPARγ would prevent corneal damage in a rat corneal alkali burn model. The authors perform a series of studies, which suggest that the combination prevents corneal inflammation and neovascularization. However, it is difficult to judge whether the study answer additive synergy of PPARα and PPARγ. The combination treatment is obviously successful in anterior segment photographs and histological analysis of the cornea, but any markers in your study would not show differences between the combination and each monotherapy. Thus, it is not possible to recommend the combination over PPAR monotherapy because enhanced monotherapy might be more effective. Please try to make the hypothesis clear and show the additive effect of the PPARs.

→You have raised a very important point of this study. In accordance with your suggestion, we conducted additional experiments to more clearly demonstrate the effects of the combination of PPARα and PPARγ. Please see the attachment. We focused on the corneal opacity which you pointed out as the effective changes of the combination drug. The opacity of the cornea results from irregular formation of the extracellular matrix in which collagen type III plays an important role. Therefore, immunohistochemical analysis of collagen type III was performed to evaluate fibrotic changes at 7days after the alkali injury (p. 9, lines 216-225; p.10, Figure 10). All PPARs treatment significantly decreased area of collagen type III compared with vehicle treatment (p.10, Figure 10e). Interestingly, percentages of collagen type Ⅲ in the corneal regions were significantly lower in the PPARs combination group compared to either PPAR alone group. We also examined the cause of strong anti-fibrotic effect of the combination. To evaluate mechanism of anti-inflammation effects, real-time RT-PCR analysis of IκB-α at 6 hours and immunohistochemical analysis of NF-κB and IκB-α at 6 hours and 7days were performed (p.4, lines 102-122; p.6, Figure 6). Instillation of PPARγ and PPARα+γ agonists reduced the mRNA expression of NF-κB in comparison with vehicle instillation. On the other hand, only PPARα treatment increased the mRNA expression of IκB-α compared with vehicle treatment. NF-κB and IκB-α stainings were observed at the similar same position in the corneal epithelium. IκB-α expression was significantly enhanced by the PPARα instillation at 6hours (p.6, Figure 6d). In the PPARγ and PPARα+γ treatment groups, there were less NF-κB expression (p.6, Figure 6e and g). On 4days in the PPARα group, NF-κB was mainly expressed in the cytoplasm and IκB-α expression was localized in the nucleus area (p.6, Figure 6k and l). The PPARγ instillation reduced NF-κB expression and there were little IκB expression in comparison with vehicle and PPARα (p.6, Figure 6m and n). The PPARα+γ treatment strongly decreased NF-κB expression compared to the other three groups (p.6, Figure 6o and p). Thus, it was suggested that different mechanism involved in NF-κB and IκB-α expression by fenofibrate and pioglitazone may contribute to the significant downregulation of NF-κB after combination of those, thereby suppressing scar formation. Combination of these PPARs may be more effective for corneal alkali burn than a single PPAR agonist.

Once again, thank you for all of the helpful suggestions for further revising our manuscript. We would like to thank you in advance for reconsidering our manuscript for publication in your journal. If you have any further questions or require other information, please feel free to contact us.

Sincerely,

Takeshi Arima, MD

Department of Ophthalmology, Nippon Medical School

1-1-5 Sendagi, Bunkyo-ku, Tokyo 113-8603, Japan

Reviewer 2 Report

Nakano and collaborator have presented an original paper entitled “Combination of peroxisome proliferator-activated receptor (PPAR) alpha and gamma agonists prevent corneal inflammation and neovascularisation in a rat alkali burn model”.

In this study, the authors wanted to verify the therapeutic impact of PPAR-a and/or PPAR-g agonists in a rat corneal alkali burn model. Ophthalmic solutions containing specific nuclear receptor agonists can suppress inflammatory cells and neovascularisation, showing therapeutic effects in corneal wound healing, combining the characteristics of both PPAR-a and PPAR-g agonists. However, the specific effect of PPAR-g agonist upregulating M2 macrophages, and PPAR-g agonist that can suppress immature blood vessels in the early phase, were evaluated. In general, studies that highlight new therapeutic options are of interest.

Every single section of the paper (introduction, material e methods, statistical errors, results, discussion) are clear and well described, but below are the specific suggestions/questions on the experimental data/analysis.

  • The authors show in figure 1a the image of the photograph of the anterior segment of the cornea after alkaline burns. It could be useful to indicate, in the figure legend or the text, when the photo was taken, if immediately after the damage or after a predetermined time.

  • Try to standardise statistics in the graphs in the figures, or remove the asterisks or add them to the missing ones (as in figure 4e or 6e, f)

  • In the materials and methods section, it would be necessary to insert the method cell count quantifying that are presented in the graphs in the paper (4 e, 6 c and f, 7 c and f).

  • The authors indicate in the discussion that the PPAR-α agonists may have a different mechanism of action than PPAR-g agonists to induce anti-inflammatory effects why can increase IkB level which inhibits the NF-kB by blocking NF-kB binding to DNA. I was hoping you could pay attention to this statement because also PPAR-g agonists have this peculiarity being able to increase Ikb and therefore, to reduce inflammation. Several papers indicate this (see some example: Scirpo, R. et al. Hepatology 62,5 (2015): 1551-62; Consoli A, Devangelio E. Lupus. 2005;14(9):794-797; Ajmone-Cat MA et al. Pharmaceuticals. 2010;3(6):1949-1965).

  • Did the authors, by chance, evaluate the effects of other molecules (e.g. caspase or ERK1/2) typically activated in this type of corneal damage? Have they been able to verify whether, even in this case, the agonists of the two PPAR isoforms can modulate the expression of these molecules? Or more, have they had the opportunity to verify, in addition to the effects on macrophages/microglia present, the consequences on astrogliosis?

  • Also, minor language improvements might be made throughout the text.

Author Response

We are grateful for the opportunity to revise and resubmit our paper (ijms-849803) entitled, “Combination of peroxisome proliferator-activated receptor (PPAR) alpha and gamma agonists prevents corneal inflammation and neovascularization in a rat alkali burn model”, and would like to thank you for your helpful comments.

Attached to this letter is a revised version of our manuscript that shows the tracked changes. In addition, we have separately listed our point-by-point responses to the comments in the text that follows.

We would like to thank you for your constructive comments. After incorporating the suggested changes, this has greatly improved our paper, thereby making this a much stronger manuscript. Please convey our thanks to you for all of your helpful input.

Our responses to the comments are as follows:

Nakano and collaborator have presented an original paper entitled “Combination of peroxisome proliferator-activated receptor (PPAR) alpha and gamma agonists prevent corneal inflammation and neovascularisation in a rat alkali burn model”.

In this study, the authors wanted to verify the therapeutic impact of PPAR-a and/or PPAR-g agonists in a rat corneal alkali burn model. Ophthalmic solutions containing specific nuclear receptor agonists can suppress inflammatory cells and neovascularisation, showing therapeutic effects in corneal wound healing, combining the characteristics of both PPAR-a and PPAR-g agonists. However, the specific effect of PPAR-g agonist upregulating M2 macrophages, and PPAR-g agonist that can suppress immature blood vessels in the early phase, were evaluated. In general, studies that highlight new therapeutic options are of interest.

→ Thank you very much for your interest in our paper’s purpose.

Every single section of the paper (introduction, material e methods, statistical errors, results, discussion) are clear and well described, but below are the specific suggestions/questions on the experimental data/analysis.

→ We really appreciate your kind comment.

The authors show in figure 1a the image of the photograph of the anterior segment of the cornea after alkaline burns. It could be useful to indicate, in the figure legend or the text, when the photo was taken, if immediately after the damage or after a predetermined time.

→ This images (p.2, Figure 1a) obtained immediately after the injury. We have revised the text (p. 2, lines 56), the figure(p. 2, Figure 1a), and the figure legend (p. 2, line 69).

Try to standardize statistics in the graphs in the figures, or remove the asterisks or add them to the missing ones (as in figure 4e or 6e, f)

→ We have revised the figures (p. 5, Figure 4e; p. 7, Figure 7c and f). We have included a new Figure (p. 6, Figure 6), so Figure 7 indicates Figure 6 of the former paper.

In the materials and methods section, it would be necessary to insert the method cell count quantifying that are presented in the graphs in the paper (4 e, 6 c and f, 7 c and f).

→ We have added the method of cell count (p. 13, line 374-375).

The authors indicate in the discussion that the PPAR-α agonists may have a different mechanism of action than PPAR-g agonists to induce anti-inflammatory effects why can increase IkB level which inhibits the NF-kB by blocking NF-kB binding to DNA. I was hoping you could pay attention to this statement because also PPAR-g agonists have this peculiarity being able to increase Ikb and therefore, to reduce inflammation. Several papers indicate this (see some example: Scirpo, R. et al. Hepatology 62,5 (2015): 1551-62; Consoli A, Devangelio E. Lupus. 2005;14(9):794-797; Ajmone-Cat MA et al. Pharmaceuticals. 2010;3(6):1949-1965).

→ Thank you for providing these insights. We have revised the text (p. 11, line 260-262), and we have included the papers you have suggested in the bibliography. Furthermore, we have performed additional experiments about NF-κB and IκΒ. Real-time RT-PCR analysis of IκB-α at 6 hours and immunohistochemical analysis of NF-κB and IκB-α at 6 hours and 7 days were performed (p.4, lines 102-122; p.6, Figure 6). Only PPARα treatment increased the mRNA expression of IκB-α compared with vehicle treatment. NF-κB and IκB-α stainings were observed at the similar same position in the corneal epithelium. IκB-α expression was significantly enhanced by the PPARα instillation at 6hours (p.6, Figure 6d). In the PPARγ and PPARα+γ treatment groups, there were less NF-κB expression (p.6, Figure 6e and g). On 4days, the PPARγ instillation reduced NF-κB expression and there were little IκB expression in comparison with vehicle and PPARα (p.6, Figure 6m and n). We have no data to show that PPARγ upregulated IκB. We would like to treat that as the subject of our analysis from now on.

Did the authors, by chance, evaluate the effects of other molecules (e.g. caspase or ERK1/2) typically activated in this type of corneal damage? Have they been able to verify whether, even in this case, the agonists of the two PPAR isoforms can modulate the expression of these molecules? Or more, have they had the opportunity to verify, in addition to the effects on macrophages/microglia present, the consequences on astrogliosis?

→ This is an interesting perspective. We have performed additional immunohistochemical analysis of caspase 3 and glial fibrillary acidic protein (GFAP), which is the main constituent of intermediate filaments in astrocytes. But there were too slight positive cells about caspase 3 to evaluate the characteristic and there were no positive areas even in the retina and optic nerve about GFAP staining. Possible reasons for the bad staining are low concentration of primary antibodies or too old antibody we used. In addition, we don’t have ERK1/2 antibody and order of the antibody cannot meet the deadline. We will make further investigation, if necessary.

Also, minor language improvements might be made throughout the text.

→ We have already gotten the English in our essay proofread. We are currently inquiring with the person in charge.

Once again, thank you for all of the helpful suggestions for further revising our manuscript. We would like to thank you in advance for reconsidering our manuscript for publication in your journal. If you have any further questions or require other information, please feel free to contact us.

Sincerely,

Takeshi Arima, MD

Department of Ophthalmology, Nippon Medical School

1-1-5 Sendagi, Bunkyo-ku, Tokyo 113-8603, Japan

Round 2

Reviewer 1 Report

Thank you for understanding my points. I would like you to continue your works for developing the field of PPARs.

Reviewer 2 Report

I thank the authors who responded promptly to all requests. In this way, they rendered the paper more complete and more attractive.